# Whole Brain Hemodynamic Response Based on Synchrony Analysis of Brain Signals for Effective Application of HD-tDCS in Stroke Patients: An fNIRS Study

**DOI:** 10.3390/jpm12030432

**Published:** 2022-03-10

**Authors:** Gihyoun Lee, Jungsoo Lee, Jinuk Kim, Heegoo Kim, Won Hyuk Chang, Yun-Hee Kim

**Affiliations:** 1Department of Health Sciences and Technology, The Samsung Advanced Institute for Health Sciences & Technology (SAIHST), Sungkyunkwan University, Seoul 06351, Korea; gihyounlee@gmail.com (G.L.); kimjuk92@gmail.com (J.K.); hiheegoo@gmail.com (H.K.); 2Department of Physical and Rehabilitation Medicine, Center for Prevention and Rehabilitation, Heart Vascular Stroke Institute, Samsung Medical Center, Sungkyunkwan University School of Medicine, Seoul 06351, Korea; wh.chang@samsung.com; 3Department of Medical IT Convergence Engineering, Kumoh National Institute of Technology, Gumi 39177, Korea; jungsoo0319@gmail.com; 4Department of Medical Device Management & Research, Department of Digital Health, The Samsung Advanced Institute for Health Sciences & Technology (SAIHST), Sungkyunkwan University, Seoul 06351, Korea

**Keywords:** hemodynamic response, brain signal synchronization, HD-tDCS, fNIRS, stroke patients

## Abstract

In this study, the effective application of high-definition transcranial direct current stimulation (HD-tDCS) based on the whole brain hemodynamic response in stroke patients was investigated using functional near-infrared spectroscopy (fNIRS). The intrahemispheric and interhemispheric synchronization and cortical activity based on the time during 1 mA HD-tDCS were examined in 26 chronic cerebrovascular disease patients. At the beginning of HD-tDCS, the synchronization and brain activity in the whole brain increased rapidly and decreased after 5 min. In the middle of tDCS, the synchronization began to increase again, and strong synchronic connections were formed around the desired stimulation area. After tDCS, strong cortical activation was observed in the stimulation area, indicating that the baseline of the oxyhemoglobin (HbO) signal increased in the desired stimulation area. Therefore, the results of this study indicate that HD-tDCS can be applied efficiently to enhance the effect of tDCS. This stimulation method with tDCS can be explored clinically for more neurorehabilitation of patients with degenerative brain diseases.

## 1. Introduction

Neuronal plasticity is one of the key factors influencing motor recovery after stroke [1]. Numerous stroke rehabilitation schemes have been developed to promote neuronal plasticity, but few among such treatments have sufficiently improved motor functions [2,3]. Transcranial direct current stimulation (tDCS) is a technique used to deliver small amounts of electric current to modulate the excitability of neural populations in different regions of the brain [4]. Recently, tDCS has been used increasingly in neuroscience and neurorehabilitation research as a non-invasive neurostimulation method to modulate neuronal cells [5]. In numerous previous studies, the effects of tDCS have shown performance gains in various cognitive [6,7] and motor domains [8,9,10] of healthy individuals. In addition, application of tDCS has shown beneficial effects in patients with chronic pain syndromes [11,12] and neuropsychiatric conditions [13,14,15] because it is well tolerated, safe, and inexpensive compared with other techniques involving invasive stimulation based on nerve signal information [16,17,18]. However, results from previous studies are inconclusive; in some studies, observed effects were not corroborated [19], and tDCS was used to investigate the polarity-specific effects that were not limited to the stimulated site [20,21,22]. Current knowledge of tDCS-induced neural changes stems from animal studies in which surface-positive current was observed to enhance neuronal firing and the size of evoked potentials [23]. Many results of previous studies indicate that tDCS induces a network and activation changes in the brain, detailed effects of tDCS based on the time and effective method to improve rehabilitation treatments are less well known. This is because the neuroimaging equipment used in previous studies is an offline method that cannot measure during tDCS, cannot observe the whole brain, or prohibits synchrony analysis because of low temporal resolution. Recently, high-definition (HD)-tDCS, which improved tDCS, has been widely used. Conventional tDCS uses saline sponge-based rectangular pads [24] which stimulates in the considerably large scalp area, but it is difficult to be combined with probe-based neuroimaging tools such as electroencephalography (EEG) and functional near-infrared spectroscopy (fNIRS) [25]. HD-tDCS contains five small ring-based electrodes which are approximately the same size as standard EEG electrodes or fNIRS optodes. They can produce focused stimulation and be simultaneously integrated with fNIRS system [25,26].

fNIRS is a method used to measure hemodynamic brain signals based on absorption of near-infrared light, with wavelengths in the range of 650–950 nm transmitted through the intact skull [27]. fNIRS monitors variations in regional cerebral blood flow and estimates hemodynamic signals highly correlated to the blood-oxygenation-level-dependent (BOLD) signal outputs in functional magnetic resonance imaging (fMRI) [28]. fNIRS is inexpensive, portable, and has the potential to extend research to a wider range of environments than many other neuroimaging systems. The advantages of fNIRS due to its near-infrared light-based neuroimaging equipment include higher temporal resolution than fMRI or positron emission tomography (PET) and stronger motion artifacts free from electrical noise and stimulation from tDCS. Several popular tools are available to analyze visually fNIRS signals such as near-infrared spectroscopy statistical parametric mapping (NIRS–SPM) [29] from Korea Advanced Institute of Science and Technology (KAIST) and *OptoNet*^®^ from Samsung Medical Center. NIRS–SPM applies the SPM method, which refers to construction and assessment of spatially extended statistical processes used to test hypotheses regarding functional imaging data [29,30]. *OptoNet*^®^ can analyze visually functional brain networks using correlation [31,32], coherence [33], frequency ratio [34], and phase-locking value (PLV) [35] based on fNIRS signals and can be utilized easily by unskilled users [36,37].

In the present study, we hypothesized that the effects of HD-tDCS would excite/depress hemodynamic responses and the resulting synchronic changes based on the time in the stimulated area as well as in other areas. The measured fNIRS signal during tDCS was divided into and analyzed as 1.25 min segments for each timeframe to investigate hemodynamic responses over time, and we analyzed the effect of HD-tDCS over time using brain activity and synchrony analysis. To the best of our knowledge, this is the first fNIRS study in which the effects of HD-tDCS on the whole brain were explored based on stimulation time. Consequently, the effective tDCS stimulation time and method were presented to enhance the stimulation effect on stroke patients using synchrony and brain network analysis using fNIRS signals.

## 2. Materials and Methods

### 2.1. Participants

To evaluate the effects of HD-tDCS using fNIRS, 26 chronic cerebrovascular disease patients who have lesions in the subcortical area (mean ± standard deviation, age: 59.4 ± 12.8 years, duration of onset: 40.1 ± 10.2 months) participated in this study. Table 1 summarizes the clinical characteristics of the participants including, Fugl-Meyer Assessment (FMA) score. The study cohort included 20 male and 6 female participants, 13 infarction and 13 hemorrhage stroke types, and 14 left- and 12 right-side stroke lesions. The experimental procedures used in this study were approved by the Institutional Review Board of Samsung Medical Center, and all participants provided written informed consent after they received a description of the study procedures and associated risks prior to the experiment.

### 2.2. Experimental Protocol

As shown in Figure 1A, the experimental protocol in this study consisted of four stages: initial rest of 0.5 min, pre-rest of 2.5 min, stimulation of 20 min, and post-rest of 5 min. The durations of ramp-up and ramp-down, which gradually increased and decreased the intensity of the tDCS at the beginning and end of the stimulation, respectively, were 30 s each. The experiment lasted a total of 28 min.

The experimental paradigm was designed to observe cortical hemodynamic responses based on time during stimulation by wearing the tDCS electrodes and the fNIRS optodes together, as shown in Figure 1B. During the initial rest, the subjects rested with their eyes closed, and the fNIRS data were collected to acquire the baseline. The subjects were asked to sit on a chair and to keep their eyes open and look at a black screen with a + mark on a monitor to avoid falling asleep during the experiment, and it was guided by the experimenter watching from the side. The fNIRS data of the pre- and post-rests were measured to compare the effect of the stimulation and was recorded continuously during tDCS. Measured fNIRS data were visualized for the whole brain to observe the effects of stimulation based on time.

### 2.3. HD-tDCS

The five tDCS electrodes were placed on the head cap as shown in Figure 1B, with the anode electrode on C4, which is the area near M1 that was connected to the cathode electrode at front, back, left, and right. The stimulation was delivered by battery driven Starstim tDCS system (Starstim 8, Neuroelectrics Inc., Barcelona, Spain). The five electrodes were used in an anodal HD-tDCS electrode configuration. The central anodal electrode and four return cathodal electrodes were located on C3 or C4 of the 10–20 system, which was chosen to stimulate M1 of the affected side with gel-filled electrodes. The current capacity of the stimulation through the anode was set to 1 mA, and each return electrode was configured to receive an equal amount of anodal current. DC currents were turned on/off slowly for 30 s each out of the field of view of the patients, a procedure that does not elicit perceived sensations.

### 2.4. fNIRS Data Acquisition

fNIRS data were acquired using an fNIRS brain imaging system (NIRScout 24–24, NIRx Medizintechnik GmbH, Berlin, Germany). This instrument has a light-emitting diode (LED) source with near-infrared light of 760 nm and 850 nm wavelengths. The arrangements of the optode structure and fNIRS channels are shown in Figure 2, with 20 NIR sources, 20 NIR detectors, and 66 fNIRS channels. The distance between the NIRS source and the detector was set to 3 cm (diagonal), and the sampling rate was fixed at 10 Hz. The head cap was attached to the subject’s head using an elastic band, and the hair near the area where all the optodes touched was set aside by an experienced experimenter to eliminate issues in emitting and detecting near-infrared light. Before data acquisition, the quality of the fNIRS data was maintained by evaluating the noise level using the NIRStar software (NIRx Medizintechnik GmbH Berlin, Germany). The fNIRS data of the patient with the left-side lesion was flipped to the opposite side of fNIRS channels, with the affected side on the right side as a reference so that the lesion location of all subject data could be analyzed.

### 2.5. Cortical Hemodynamic Response Analysis

In the present study, to analyze the cortical hemodynamic responses, the analysis method was conducted with cortical activity and cortical synchrony. First, the cortical activation analysis of fNIRS data was performed using the open-source toolbox NIRS-SPM implemented in MATLAB^®^ (MathWorks, Inc., Natick, MA, USA). The NIRS-SPM [29] (http://bisp.kaist.ac.kr/NIRS-SPM, 29 October 2020) is a widely used cortical activity mapping software in the fNIRS field and was modified from fMRI-SPM [38], which is used widely in the MRI field. In SPM analysis [39], the general linear model (GLM) with a canonical hemodynamic response was performed to model the estimated oxyhemoglobin (HbO) and deoxyhemoglobin (HbR) signals. Then, the statistical contrast in reference to the base signal was tested, and cortical activity was represented as t-value during the experiment. In the group analysis for all subjects, statistical analysis was performed based on the individual-level beta values to determine activated channels. Then, t-statistic maps computed for group analysis were plotted onto a conventional brain template, and the regions with significant contrast in HbO concentrations were identified.

The cortical synchrony analysis of fNIRS data was performed using the *OptoNet* II^®^ software (https://sites.google.com/site/dsucore/free/optonet, 25 March 2021), which is a MATLAB-based application for functional cortical connectivity analysis of fNIRS signals [36,37]. The software analyzes the connectivity of fNIRS channels and connectivity between the functional regions of the brain cortex. In the present study, the functional region analysis and PLV in *OptoNet* II^®^ were used to estimate the connectivity and synchrony between functional domains. The phase-locking value (PVL) can detect synchrony in a precise frequency range between two recording sites, uses responses to a repeated stimulus, and searches for latencies at which the phase difference between the signals varies minimally across trials (phase-locking) [35]. The PLV measures the intertrial variability of this phase difference; if the phase difference varies minimally across trials, PLV is close to 1, otherwise it is close to zero [35]. In addition, PLV is suitable for analyzing synchrony and synchrony of fNIRS signals, which have a lower temporal resolution than EEG signals, because it can cover a broad frequency range and time delay by repeating the PLV calculation. fNIRS channels of the cortical functional regions recommended by the fNIRS optodes location decider (fOLD) toolbox [40] in MATLAB^®^ were grouped as follows: medial pre-frontal (MPF): CH 1, 2, 3, 4, 5, 7, 8; left frontal (Lt. Fr): 6, 14, 15, 16, 18, 19; right (Rt.) Fr: 10, 11, 12, 13, 29, 30; Lt. primary motor cortex (M1): 22, 23, 24, 28, 59; Rt. M1: 34, 35, 37, 39, 43; supplementary motor area (SMA): 9, 25, 27; Lt. pre-motor cortex (PM): 17, 21, 26; Rt. PM: 31, 32, 33; Lt. temporal lobe (Tm): 20, 56, 58; Rt. Tm: 36, 38, 50; Lt. sensory cortex (Sn): 46, 53, 57 60, 61; Rt. Sn: 40, 41, 44, 45; Lt. parietal (Pr): 52, 54, 55, 62, 3, 64, 65; Rt. Pr: 42, 47, 48, 49, 51, 66; and occipital cortex (Occ): 67 (Figure 1C). To avoid signal distortion from differences in the number of fNIRS channels between functional region groups, the fNIRS signals were processed with normalization and ensemble averaging for each epoch.

## 3. Results

### 3.1. Activation Analysis Results Using NIRS-SPM

Figure 2 shows the results of time-dependent analysis of the cortical activity from the hemodynamic responses of the measured fNIRS signal using NIRS-SPM [29]. The fNIRS signal was divided into 2.5 min in reference to the fNIRS signal of the pre-rest stage for 2.5 min, and it was analyzed using the SPM analysis and presented as a brain activity map where the regions have a *p*-value < 5/100. Therefore, there was no activation in the pre-rest stage, and cortical activation is shown in the regions with significant contrast with the fNIRS signal of pre-rest stage. The activation map was presented based on t-value, if an area has a bright color (white or yellow), it means there is high activation. Finally, cortical activation maps for a total of 11 images (one pre-rest, eight during tDCS, and two post-rest) were completed as shown in Figure 2. After a few minutes of stimulation, strong cortical activation was noticed in wide brain areas such as MPF, Fr, M1, Tm, SMA, and PM, which was lasted for 5 min. The cortical activation then decreased until the tDCS ended and was not observed in almost all areas thereafter. After tDCS, strong cortical activations were observed around the area to which the stimulation was applied and lasted over 5 min.

### 3.2. Synchrony Analysis Results Using OptoNet II^®^

Figure 3 shows the results of cortical synchrony estimated using *OptoNet* II^®^. The synchrony maps showed results for 22 timeframes, which were divided into a timeframe for every 1.25 min. The synchrony line was represented by warmer color if PLV was close to 1 and colder color if PLV was close to 0, and the only high-value lines (>0.7) were represented in Figure 3. The results showed strong synchronization in all brain areas immediately after onset of tDCS and stronger synchronization on the affected side to which the stimulation was applied than the unaffected side. After starting tDCS, the number of synchronized connections with PLV over 0.7 dramatically decreased from 210 connections at the 8th timeframe to 12 connections at the 9th timeframe in the whole brain. In contrast, the number of connections related to M1 in the affected side increased from 1 to 8 between the 13th and 14th timeframe. On the other hand, there was no change in the number of connections in the unaffected side. The synchronic connection became stronger around the stimulated region until 20 min after tDCS.

To investigate synchrony based on the timeframe, the PLV in each hemisphere was analyzed numerically and statistically. Figure 4 shows the average PVL for the whole brain and the intra PLV of each hemisphere, representing the mean result only in the synchronic connection within the same hemisphere based on the timeframe, and statistically significant differences of each timeframe compared to the pre-test value were investigated using paired *t*-test. The values for the whole brain show the average value of the two hemispheres. The affected side showed a numerical higher increase of mean PLV after synchronization until the end of stimulation from the 9th to 18th timeframe as 0.3669 ± 0.1990 to 0.7894 ± 0.1790 (mean ± standard deviation) on the unaffected side and 0.3223 ± 0.1599 to 0.8331 ± 0.0979 on the affected side, respectively. The difference between the 9th and 18th timeframes on each hemisphere was 0.5535 ± 0.1196 on the affected side and 0.3831 ± 0.2139 on the unaffected side which showed marginal significance (*p* = 5.3/100). Furthermore, mean PLV values showed statistically significant differences between the two hemispheres at the 19th and 20th timeframes after the end of tDCS. Apparently, the stimulation applied to the affected side was transmitted to the unaffected side, and the rationale for transmission of stimulation effects is demonstrated in Figure 5. Figure 5 only shows the number of all synchrony connections, which are was represented as stacked bars of various colors over each threshold, except the connections that have PLVs under 0.5. To estimate the significant differences of each timeframe compared to the pre-rest, the results of PLV were investigated using Wilcoxon signed-rank test by converting continuous values into ordinal values (*n_o_*, 0 ≤ PLV < 0.5∶ *n_o_* = 0; 0.5 ≤ PLV < 0.6∶ *n_o_* = 1; 0.6 ≤ PLV < 0.7∶ *n_o_* = 2; 0.7 ≤ PLV < 0.8∶ *n_o_* = 3; 0.8 ≤ PLV < 0.9: *n_o_* = 4; 0.9 ≤ PLV < 0.95: *n_o_* = 5; 0.95 ≤ PLV ≤ 1: *n_o_* = 6). In the inter synchrony connection between the two hemispheres, as shown in Figure 5A and similar to the results in Figure 4, the number of synchronies above each threshold increased or decreased at certain rates. Conversely, in Figure 5B,C showing the number of synchronic connections in each hemisphere, the results with the highest synchronization (the sky-blue color bar, PLV > 0.95) showed numerous differences. In timeframes 3–5, which was the beginning of tDCS, most of the synchrony connections have the PLVs 0.95 on the affected side, and this effect was transmitted to the unaffected side, resulting in higher values than the affected side in the 7th and 8th timeframes. After the second half and end of tDCS, the stimulation was effectively delivered, showing high numbers of synchrony and significant differences in the affected side. The affected side was more strongly excited due to the response to tDCS, decreased more due to the inverse effect, and recovered to a higher level than that of the unaffected side. This phenomenon indicates that the strong whole brain activation and high synchronization in the early stage of tDCS are from the brain nerve excitation caused by sudden stimulation and confirmed that it was transmitted from the stimulated side to the opposite side over time.

## 4. Discussion

In all previous studies on the effects of tDCS, an increase of HbO was reported; however, brain activation of unstimulated areas was reported in some studies [41,42,43]. It was necessary to confirm whether this phenomenon was stimulated by tDCS and resulted in increased blood flow or an excitatory response of the whole brain due to neurological excitation due to sudden stimulation. Therefore, in the present study, fNIRS signals of high temporal resolution in the whole brain during tDCS were measured, and cortical activity and synchrony simultaneously were analyzed based on time. Strong cortical activity and synchronization in the early stage of tDCS were shown in most of the brain, followed by a sharp decrease in cortical activity and synchrony, which was reported in several previous studies [44,45].

The strong synchronization and increasing HbO in the early stage (approximate timeframe 3rd–8th, 0–7.5 min after the start of tDCS) are from the excitation of the neural nerves by tDCS, which was confirmed by synchrony analysis in this study. Subsequently, in the intermediate stage (approximate timeframe 8th–12th, 7.5–12.5 min after the start of tDCS), both HbO and synchronic connections rapidly decreased. This can be explained by hemodynamic redistribution of blood flow after stimuli excitation and altered neuronal transmission, which were reported in previous studies [42,43,46]. Then, until the end of tDCS (approximate timeframe 12th–18th, 12–20 min after the start of tDCS), the synchronic connections continued to increase around the stimulated area, so the most efficient stimulation was applied to the desired region in this stage. After stimulation (approximate timeframe 19th–22nd, 20–25 min after the start of tDCS), cortical activation stopped decreasing and was observed again around the stimulated area, which increased the HbO baseline and efficiency of functional exercise. Although studies on patients showed improvement in various symptoms, tDCS effects were less consistent, similar to the effects on clinical populations and their associated symptoms [47]. Therefore, in the present study, a method to enhance the stimulation effect of chronic cerebrovascular disease patients through whole brain activity and synchronization analysis was proposed. Based on cortical activation and synchrony analysis, the results of the synchrony and activation maps showed similar trends. In addition, the synchrony analysis result was correlated strongly with the hemodynamic response to tDCS because synchronization was affected more sensitively by stimulation and recovered faster. Consequently, tDCS stimulation within 12 min did not properly stimulate the desired region for stroke patients, indicating that a significant stimulation effect is unlikely, and tDCS can be applied effectively to the desired regions when stimulated for more than 15 min. In addition, sufficient rehabilitation exercises for stroke patients are effective with cortical activation in the stimulated regions after tDCS.

## 5. Conclusions

In the present study, the effects of HD-tDCS on whole brain hemodynamic responses based on time were demonstrated using cortical activation and synchrony analysis. The results showed that cortical activity was increased at the beginning of tDCS, continually decreased until tDCS was ended, and was maintained at a high level around the stimulated area after stimulation was finished. Synchrony analysis showed a very strong synchronization at the start of tDCS, then a sharp decrease after 5 min, and a continuous increase in the synchronic connections around the stimulated area from 12 min until tDCS was ended. Based on the analysis results, the minimum duration for effective stimulation of tDCS in the desired area was proposed, and a time point for effective rehabilitation exercise was suggested. Using the results of this study, it is possible to improve the treatment effects for ischemic stroke survivors [48,49] poststroke depression [50], schizophrenia [51], nicotine dependence [52], tinnitus [53], and various motor functions which are currently being attempted by combining tDCS and fNIRS. In addition, combining HD-tDCS, which provides focused stimulation, is expected to be utilized for more neurologic diseases. However, only one neurologic population participated in this study, and it needs to apply and confirm not only to other neurologic populations but also to healthy people. In the future, we can extend this study to confirm the effects of rehabilitation for various neurologic diseases. Finally, our findings can be further enhanced and utilized in clinical procedures to treat patients with various other types of neurologic diseases as well as to improve the brain function of healthy people.

## Figures and Tables

**Figure 1 jpm-12-00432-f001:**
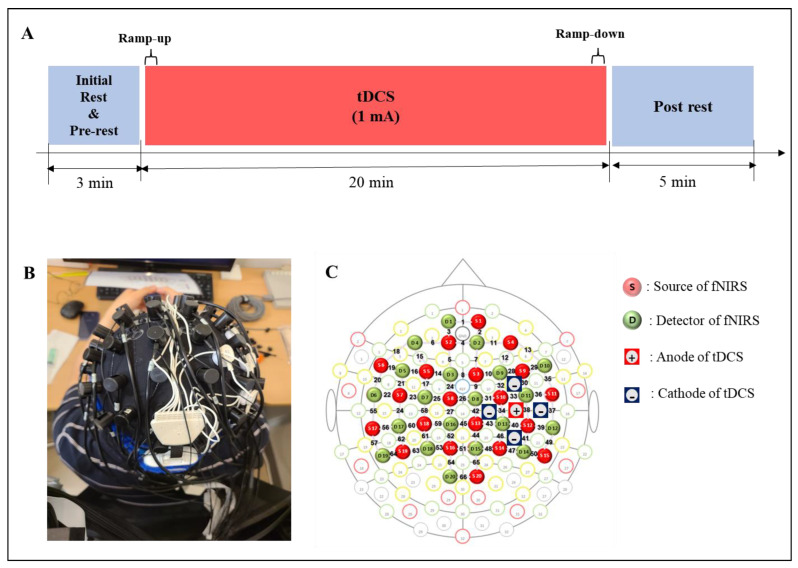
Experimental design. (**A**) Experimental conditions. (**B**) A head cap including the tDCS electrodes and fNIRS optodes. (**C**) Topography for tDCS electrodes and fNIRS optodes location.

**Figure 2 jpm-12-00432-f002:**
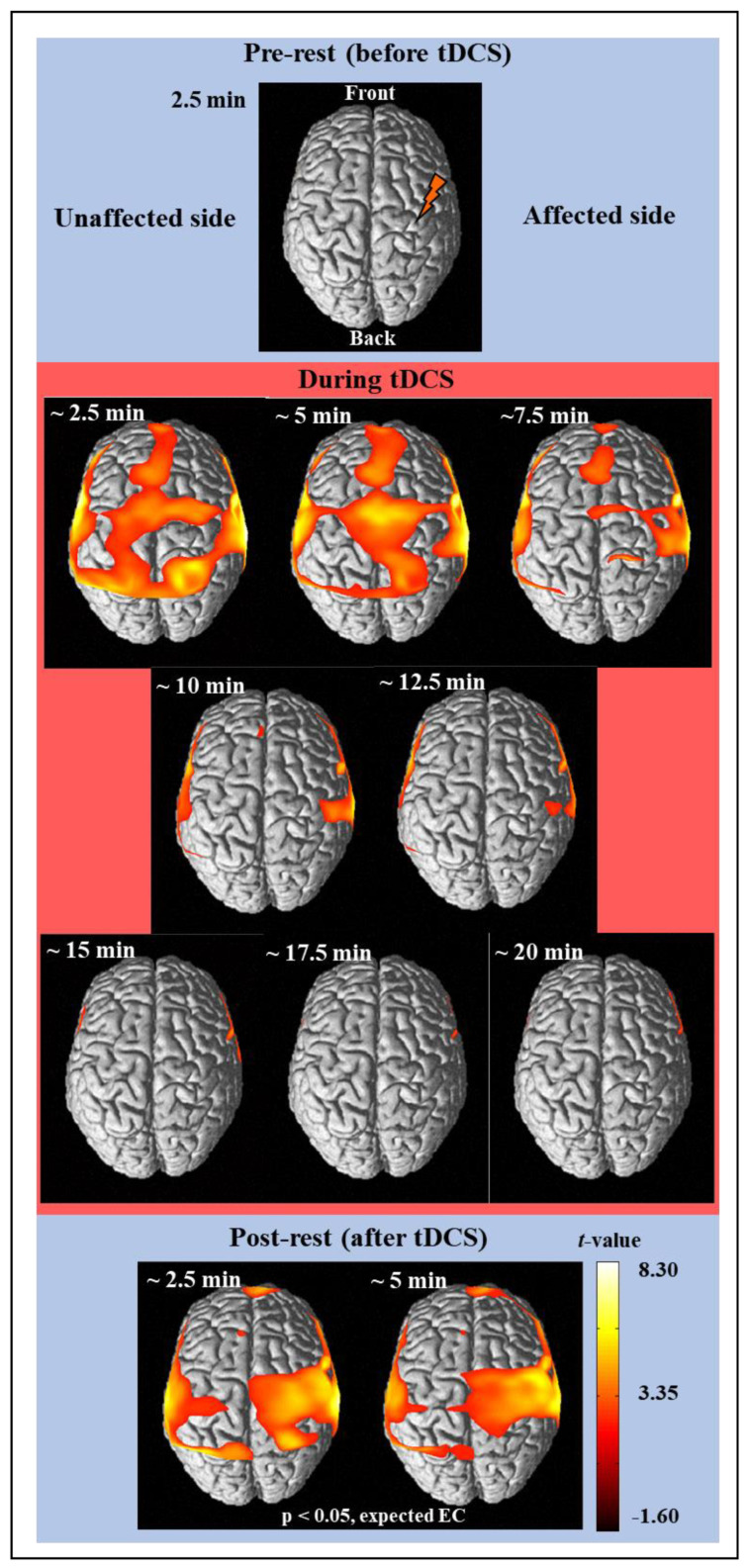
The average cortical activation maps of 26 subjects analyzed by using the NIRS-SPM software before (prerest), during, and after (post-rest) tDCS intervention.

**Figure 3 jpm-12-00432-f003:**
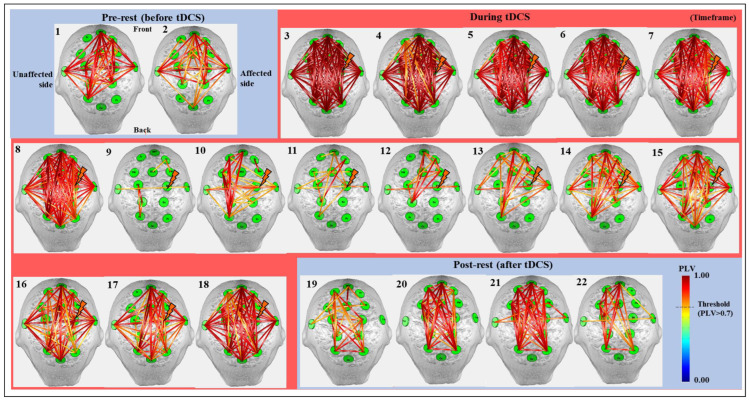
The cortical synchrony results analyzed by using the *OptoNet* II^®^ software based on the total 22 timeframes in all 26 subjects. The green circles—functional brain regions, the colored lines—the synchrony connections, the lightning mark—the tDCS target.

**Figure 4 jpm-12-00432-f004:**
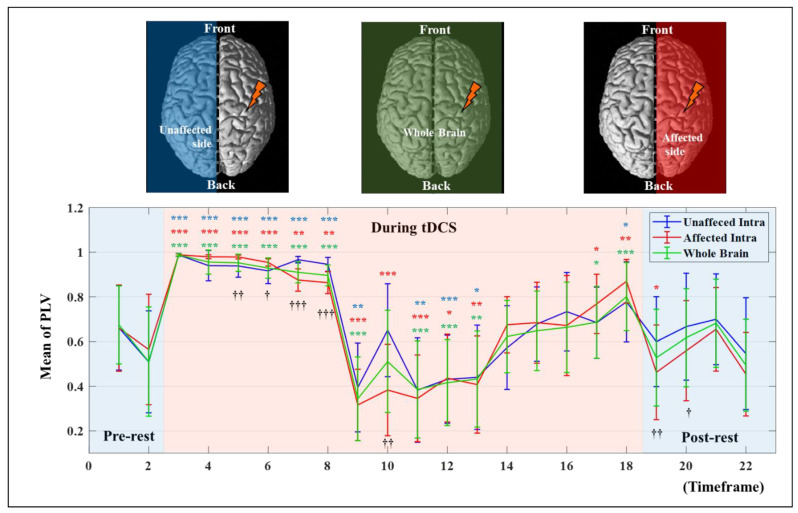
The mean PLV values based on the timeframe in each hemisphere and a whole brain. The error bar indicates the standard deviations between all related channels. The asterisk indicates a statistical difference compared to the pre-test value (* *p* < 5/100; ** *p* < 1/100; *** *p* < 1/1000), and the cross indicates a statistical difference between affected and unaffected intrahemispheric connections († *p* < 5/100; †† *p* < 1/100; ††† *p* < 1/1000).

**Figure 5 jpm-12-00432-f005:**
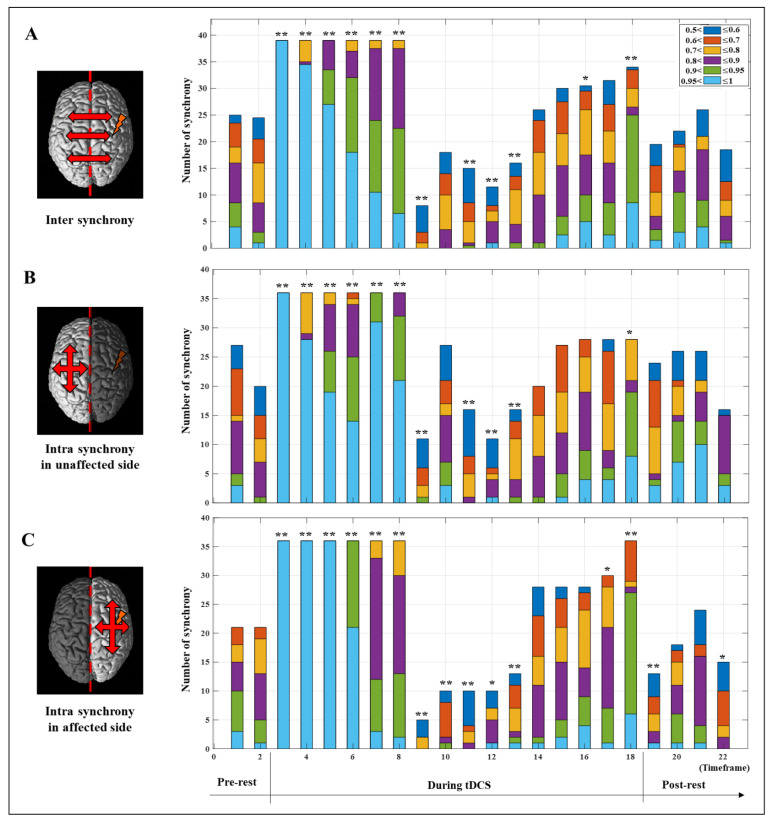
The number of synchrony connections over the threshold level before (pre-test), during, and after (post-rest) tDCS intervention. (**A**) Inter synchrony showed number of synchrony connections between the two hemispheres. Intra synchrony showed number of synchrony connections within the unaffected hemisphere (**B**) and the affected hemisphere (**C**). The asterisk indicates a statistical difference compared to the pre-test value (* *p* < /1005; ** *p* < 1/100)).

**Table 1 jpm-12-00432-t001:** The clinical characteristics of participants.

Characteristics	*n* = 26
Age (years)	
Mean ± SD	59.4 ± 12.8
Sex (*n*)	
Male	20
Female	6
Stroke type (*n*)	
Infarction	13
Hemorrhage	13
Lesion location (*n*)	
Supratentoria	
Cortical	
MCA territory (Lt./Rt.)	1 (0/1)
Subcortical	
CR (Lt./Rt.)	3 (0/3)
BG (Lt./Rt.)	10 (5/5)
CR + BG (Lt./Rt.)	1 (1/0)
Thalamus (Lt./Rt.)	5 (3/2)
Infratentorial	
Pons (Lt./Rt.)	5 (5/0)
Medullar (Lt./Rt.)	1 (0/1)
Duration (months)	
Mean ± SD	40.1 ± 29.4
FMA upper extremity score	
Mean ± SD	47.6 ± 10.2
FMA total score	
Mean ± SD	69.3 ± 14.1

SD, standard deviation; Lt., left; Rt., right; MCA, middle cerebral artery; CR, corona radiata; BG, basal ganglia; FMA, Fugl-Meyer Assessment.

## Data Availability

The data that support the findings of this study are available from the corresponding author, Y.-H.K., upon reasonable request.

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
