# Peer review of "Whole Brain Hemodynamic Response Based on Synchrony Analysis of Brain Signals for Effective Application of HD-tDCS in Stroke Patients: An fNIRS Study"

_jpm, 2022, doi:10.3390/jpm12030432_

Round 1
Reviewer 1 Report
Please, see attached file

Author Response
Response to Reviews
We sincerely thank the reviewers for their constructive and valuable comments. We have addressed their specific concerns below and have made the suggested changes which were very helpful in improving the manuscript. Point-by-point responses to the reviewers’ comments are written in blue as follows.
Comments and Suggestions for Authors
- In the hemodynamic response analysis, results are based on HbO signals. Are there any results using the HbR signals?
- We also have HbR signals from fNIRS. The reason why we present all results based on HbO signal is that HbO and HbR basically have a similar trend and HbO signals have much higher CNRs (Contrast-to-Noise Ratio) and SNRs (Signal-to-Noise Ratio) than HbR signal. Sometimes, HbR signals did not show significant difference between block, channel, and time, etc, even though it has similar characteristics and features with HbO signals.
- According to your comments, we analyzed PLV results of HbR according to time-frames shown in the picture below. If you compare the results with those of HbO presented in Figure 3 of the manuscript, you may notice that the results show a similar trend to those of HbO, but less difference. Therefore, we thought that presenting HbO signals were more distinguishable to explain the contrasting results to the readers.
- The demographic characteristics of the sample used are described in Table 1. These characteristics are not used anymore subsequently. For example it would be useful to present the difference of HbO signal from the base signal at preselected time-points (e.g. during tDCS at 5 min, post tDCS at 5 min) by each covariate (categorized age, sex, stroke type, stroke location, categorized duration, categorized FMA score), accompanied by a p-value.
- We totally agree with the reviewer’s comment. However, the main purpose of this study is to observe hemodynamic response for effective application of HD-tDCS in chronic stroke based on fNIRS signal. We described the detailed clinical characteristics of the participants in Table 1 to represent that all subjects were in a chronic stage beyond 6 months after onset when natural recovery is generally over. And most of the subject’s stroke lesions were located at the subcortical or infratentorial areas, therefore, not directly influencing the cortical geometry. tDCS is known to mostly affect the cortical excitability and fNIRS also monitors the hemodynamic response on the cortical level [1, 2]. To support these facts, Table 1 has been presented.
- We also agree that further research using the clinical population you mentioned by using each covariate is needed. Therefore, we revised the conclusion including the limitations of that and the need for future work as follows.
- “In addition, combining HD-tDCS, which provides focused stimulation, is expected to be utilized for more neurologic diseases. However, only one neurologic population was conducted in this study, and it needs to apply and confirm not only to other neurologic populations but also to healthy people. In the future, we can extend this study to confirm the effects of rehabilitation for various neurologic diseases. Finally, our findings can be further enhanced and utilized in clinical procedures to treat patients with various other types of neurologic diseases as well as to improve the brain function of healthy people.”
- The t-values presented in Figure 2 are unadjusted ones. A more appropriate approach, would be to employ a model for the difference of the HbO signal from the base signal, call it diff(i,t) for subject i (i = 1,2,….26), at time t (t=1,2,… 11 - one pretest, 8 during tDCS and 2 post-test time-points). The model could take the following form:
diff(i,t) = subject (random effect) +
Covariates (e.g. age, sex, stroke type, location, duration, FMA score) +
cortical regions (e.g. MPF, M1, SMA, …) +
time (as factor) + error
Using the above model we could estimate the adjusted diff(i,t) along with its standard error, for all cortical regions at all time points and examine whether it differs from zero.
- Thank you for your kind review. To be precise, these results are adjusted t-value in Figure 2. We used HbO signal of 3 minutes before tDCS as a baseline signal. The Covariates you mentioned were not used, but the t-value was estimated using the statistical difference from the base signal for each fNIRS channel for each subject at time-frame. And after that, it was the result to be obtained by performing group analysis with normalized data for all subjects. This method has been confirmed in a lot of studies and the toolbox called fNIRS-SPM[2] that we used for that is one of the most widely used for brain activity analysis using fNIRS until recently[3-8], and we can say it satisfies what you mentioned. And we added the contents about the limitations of the clinical population and the need for future work in Conclusion.
- Similar comments apply for the synchrony analysis. Average PLV and number of synchrony should be presented by demographic information at selected time-frames. A similar model as above for average PLV and number of synchrony should be employed to assess differences of the response at various time-frames.
- We agree with your comment. However, the number of subjects was too small to represent demographic information, and it was not included because it did not fit the purpose of this study. We ask for your understanding of that and have mentioned this as a limitation and need for a future work in the Conclusion section of the revised manuscript.
- Finally, multiplicity issues should alter the value at which we declare statistical significance for the univariate comparisons. For example p<1/1000 (i.e. p<0.001) might be more appropriate than p<0.05 or p<0.01.
- We revised the statistical significance as you mentioned.
Thank you for your thoughtful review. We also revised the whole manuscript more clearly understandable to the readers. We’re so grateful for your detailed comments and advice to make our manuscript more valuable. If there is anything to be changed or revised, please let us know at your earliest convenience. We shall do our best to respond it as soon as possible.
References
[1] R. Patel, A. Dawidziuk, A. W. Darzi, H. Singh, and D. R. Leff, "Systematic review of combined functional near-infrared spectroscopy and transcranial direct-current stimulation studies," Neurophotonics, vol. 7, no. 2, p. 020901, 2020.
[2] J. C. Ye, S. Tak, K. E. Jang, J. Jung, and J. Jang, "NIRS-SPM: statistical parametric mapping for near-infrared spectroscopy," Neuroimage, vol. 44, no. 2, pp. 428-47, Jan 15 2009.
[3] G. Lee, J.-S. Park, J. Lee, J. Kim, Y.-J. Jung, and Y.-H. Kim, "OptoNet II: An Advanced MATLAB-Based Toolbox for Functional Cortical Connectivity Analysis With Surrogate Tests Using fNIRS," IEEE Access, vol. 9, pp. 15983-15991, 2020.
[4] H. Yamazaki, Y. Kanazawa, and K. Omori, "Advantages of double density alignment of fNIRS optodes to evaluate cortical activities related to phonological short-term memory using NIRS-SPM," Hearing Research, vol. 395, p. 108024, 2020.
[5] T. Watanabe, T. Fujiwara, and S. Suzuki, "NIRS-SPM analysis of body schema modification and performance of body motion," in 2018 11th International Conference on Human System Interaction (HSI), 2018, pp. 369-374: IEEE.
[6] G. Lee, J.-S. Park, and Y.-J. Jung, "OptoNet: a MATLAB-based toolbox for cortical network analyses using functional near-infrared spectroscopy," Optical Engineering, vol. 59, no. 06, 2019.
[7] G. Lee, J.-S. Park, M. L. B. Ortiz, J.-Y. Hong, S.-H. Paik, S. H. Lee, B. M. Kim, and Y.-J. Jung, "Hemodynamic Activity and Connectivity of the Prefrontal Cortex by Using Functional Near-Infrared Spectroscopy during Color-Word Interference Test in Korean and English Language," Brain Sciences, vol. 10, no. 8, p. 484, 2020.
[8] B. E. White, "Converting POS Files for Use with the NIRS-SPM and SPM-fNIRS Toolboxes via POS2SPM," 2016.

Reviewer 2 Report
-The introduction is missing a short paragraph on HD-tDCS technique and the explanation for the differences of classical tDCS vs HD-tDCS and references on the use of HD-tDCS in stroke patients. The explanation for the use of HD-tDCS in the rehabilitation of stroke patients or it is used more in research studies in stroke patients.
-When introducing the new acronym (HD-tDCS) the full term should be given for the first use in the manuscript text (excluding abstract). The same is valid for the FMA acronym.
-row 75- 76 please be more clear and update the sentence regarding the term “scientific analysis”
“ Consequently, the effective tDCS stimulation time and method were presented to enhance the stimulation effect on stroke patients using scientific analysis.”
-Please carefully check the stroke sides, in the text (row84 it says 12 left and 14 right, while in Table 1 is vice versa)?
-297 row the references are missing
-the conclusions rows 295-301 should be updated to be more clearer and concreate. It seems like the HD-tDCS is the “” ready to be used in clinics for a variety of pathologies. The authors should concentrate on the future for this neurologic population that they are studying. Also, elaborate on the differences in tDCS vs HD-tDCS regarding the study results and the practical knowledge for clinicians (mostly using tDCS).
-The authors conducted the study on one neurologic population and this should be given in the limitations of the study since the Discussion does not have a limitation paragraph. Also, the study here was not having control healthy subjects and this fact needs to be discussed.
Author Response
Response to Reviews
We sincerely thank the reviewers for their constructive and valuable comments. We have addressed their specific concerns below and have made the suggested changes which were very helpful in improving the manuscript. point-by-point responses to the reviewers’ comments are written in blue as follows.
Comments and Suggestions for Authors
- The introduction is missing a short paragraph on HD-tDCS technique and the explanation for the differences of classical tDCS vs HD-tDCS and references on the use of HD-tDCS in stroke patients. The explanation for the use of HD-tDCS in the rehabilitation of stroke patients or it is used more in research studies in stroke patients.
- We agree with reviewer’s comment. We revised the introduction as you recommended. Descriptions have been added about the differences of classical tDCS vs HD-tDCS and the use of HD-tDCS in the rehabilitation of stroke as follows:
- “Neuronal plasticity is one of the key factors influencing motor recovery after stroke [1]. Numerous stroke rehabilitation schemes have been developed to promote neuronal plasticity, but few among such treatments have sufficiently improved motor functions [2, 3].”
- “Recently, high-definition (HD)-tDCS, which improved tDCS, has been widely used. Conventional tDCS uses saline sponge-based rectangular pads [4] which stimulates in the considerably large scalp area, but it is difficult to be combined with probe based-neuroimaging tools such as electroencephalography (EEG) and functional near-infrared spectroscopy (fNIRS) [5]. HD-tDCS contains 5 small ring-based electrodes which are approximately the same size as standard EEG electrodes or fNIRS optodes. They can produce focused stimulation and be simultaneously integrated with fNIRS system [5, 6].”
- When introducing the new acronym (HD-tDCS) the full term should be given for the first use in the manuscript text (excluding abstract). The same is valid for the FMA acronym.
- We revised the acronym as you mentioned.
- row 75- 76 please be more clear and update the sentence regarding the term “scientific analysis”
“Consequently, the effective tDCS stimulation time and method were presented to enhance the stimulation effect on stroke patients using scientific analysis.”
- We agree with reviewer’s comment. We revised the term to “synchrony and brain network analysis using fNIRS signals”.
- Please carefully check the stroke sides, in the text (row84 it says 12 left and 14 right, while in Table 1 is vice versa)?
- It was our mistake, thank you for your thoughtful review. We revised the text to “14 left and 12 right stroke lesion sides”.
- 297 row the references are missing
- We filled the references as a follow.
- “Using the results of this study, it is possible to improve the treatment effects for ischemic stroke survivors [7, 8] poststroke depression [9], schizophrenia [10], nicotine dependence [11], tinnitus [12], and various motor function which are currently being attempted by combining tDCS and fNIRS.”
-the conclusions rows 295-301 should be updated to be more clearer and concreate. It seems like the HD-tDCS is the “” ready to be used in clinics for a variety of pathologies. The authors should concentrate on the future for this neurologic population that they are studying. Also, elaborate on the differences in tDCS vs HD-tDCS regarding the study results and the practical knowledge for clinicians (mostly using tDCS).
- We agree with the reviewer’s comment. We revised the Conclusions adding the content about applying to the other neurologic diseases, and the advantages from the differences in tDCS and HD-tDCS as follows.
- “In addition, combining HD-tDCS, which provides focused stimulation, is expected to be utilized for more neurologic diseases.”
- “In the future, we can extend this study to confirm the effects of rehabilitation for various neurologic diseases. Finally, our findings can be further enhanced and utilized in clinical procedures to treat patients with various other types of neurologic diseases as well as to improve the brain function of healthy people.”
-The authors conducted the study on one neurologic population and this should be given in the limitations of the study since the Discussion does not have a limitation paragraph. Also, the study here was not having control healthy subjects and this fact needs to be discussed.
- We agree with the reviewer’s comment. We added the limitations of the fact that conducted only one the neurologic population, did not have control healthy subjects, and need to further work in Conclusion as a follow:
- “However, only one neurologic population was conducted in this study, and it needs to apply and confirm not only to other neurologic populations but also to healthy people.”
Thank you for your thoughtful review. We also revised the whole manuscript more clearly understandable to the readers. We’re so grateful for your detailed comments and advice to make our manuscript more valuable. If there is anything to be changed or revised, please let us know at your earliest convenience. We shall do our best to respond it as soon as possible.
References
[1] S. Li, "Spasticity, motor recovery, and neural plasticity after stroke," Frontiers in neurology, vol. 8, p. 120, 2017.
[2] M. Iosa, G. Morone, A. Fusco, M. Bragoni, P. Coiro, M. Multari, V. Venturiero, D. De Angelis, L. Pratesi, and S. Paolucci, "Seven capital devices for the future of stroke rehabilitation," Stroke research and treatment, vol. 2012, 2012.
[3] P. Langhorne, F. Coupar, and A. Pollock, "Motor recovery after stroke: a systematic review," The Lancet Neurology, vol. 8, no. 8, pp. 741-754, 2009.
[4] A. R. Brunoni, M. A. Nitsche, N. Bolognini, M. Bikson, T. Wagner, L. Merabet, D. J. Edwards, A. Valero-Cabre, A. Rotenberg, and A. Pascual-Leone, "Clinical research with transcranial direct current stimulation (tDCS): challenges and future directions," Brain stimulation, vol. 5, no. 3, pp. 175-195, 2012.
[5] S.-C. Bao, W.-W. Wong, T. W. H. Leung, and K.-Y. Tong, "Cortico-muscular coherence modulated by high-definition transcranial direct current stimulation in people with chronic stroke," IEEE Transactions on Neural Systems and Rehabilitation Engineering, vol. 27, no. 2, pp. 304-313, 2018.
[6] S. Nikolin, C. K. Loo, S. Bai, S. Dokos, and D. M. Martin, "Focalised stimulation using high definition transcranial direct current stimulation (HD-tDCS) to investigate declarative verbal learning and memory functioning," Neuroimage, vol. 117, pp. 11-19, 2015.
[7] A. Dutta, A. Jacob, S. R. Chowdhury, A. Das, and M. A. Nitsche, "EEG-NIRS based assessment of neurovascular coupling during anodal transcranial direct current stimulation-a stroke case series," Journal of medical systems, vol. 39, no. 4, pp. 1-9, 2015.
[8] U. Jindal, M. Sood, S. R. Chowdhury, A. Das, D. Kondziella, and A. Dutta, "Corticospinal excitability changes to anodal tDCS elucidated with NIRS-EEG joint-imaging: An ischemic stroke study," in 2015 37th Annual International Conference of the IEEE Engineering in Medicine and Biology Society (EMBC), 2015, pp. 3399-3402: IEEE.
[9] H. Li, N. Zhu, E. A. Klomparens, S. Xu, M. Wang, Q. Wang, J. Wang, and L. Song, "Application of functional near-infrared spectroscopy to explore the neural mechanism of transcranial direct current stimulation for post-stroke depression," Neurological Research, vol. 41, no. 8, pp. 714-721, 2019.
[10] Z. Narita, T. Noda, S. Setoyama, K. Sueyoshi, T. Inagawa, and T. Sumiyoshi, "The effect of transcranial direct current stimulation on psychotic symptoms of schizophrenia is associated with oxy-hemoglobin concentrations in the brain as measured by near-infrared spectroscopy: a pilot study," Journal of Psychiatric Research, vol. 103, pp. 5-9, 2018.
[11] I. A. Dias, F. A. Hazime, D. A. Lopes, C. S. da Silva, A. F. Baptista, and B. A. K. da Silva, "Effects of transcranial direct current stimulation on heart rate variability: a systematic review protocol," JBI evidence synthesis, vol. 18, no. 6, pp. 1313-1319, 2020.
[12] R. Verma, A. Jha, and S. Singh, "Functional near-infrared spectroscopy to probe tDCS-induced cortical functioning changes in tinnitus," The Journal of International Advanced Otology, vol. 15, no. 2, p. 321, 2019.

Round 2
Reviewer 1 Report
The author's replies have addressed the issues put forward after the 1st review round. It is recommended to aceept it for publication
Reviewer 2 Report
I thank the authors for the responses they answered all concernes.